# A Preliminary Impact Study of Wind on Assimilation and Forecast Systems into the One-Dimensional Fog Forecasting Model COBEL-ISBA over Morocco

**Driss Bari** 

Direction de la Météorologie Nationale, Casablanca 22000, Morocco; bari.driss@gmail.com; Tel.: +212-6600-78079

**Abstract:** The assimilation impact of wind data from aircraft measurements (AMDAR), surface synoptic observations (SYNOP) and 3D numerical weather prediction (NWP) mesoscale model, on short-range numerical weather forecasting (up to 12 h) and on the assimilation system, using the one-dimensional fog forecasting model COBEL-ISBA (Code de Brouillard à l'Échelle Locale-Interactions Soil Biosphere Atmosphere), is studied in the present work. The wind data are extracted at Nouasseur airport, Casablanca, Morocco, over a winter period from the national meteorological database. It is the first time that wind profiles (up to 1300 m) are assimilated in the framework of a single-column model. The impact is assessed by performing NWP experiments with data denial tests, configured to be close to the operational settings. The assimilation system estimates the flow-dependent background covariances for each run of the model and takes the cross-correlations between temperature, humidity and wind components into account. When assimilated into COBEL-ISBA with an hourly update cycle, the wind field has a positive impact on temperature and specific humidity analysis and forecasts accuracy. Thus, a superior fit of the analysis background fields to observations is found when assimilating AMDAR without NWP wind data. The latter has shown a detrimental impact in all experiments. Besides, wind assimilation gave a clear improvement to short-range forecasts of near-surface thermodynamical parameters. Although, assimilation of SYNOP and AMDAR wind measurements slightly improves the probability of detection of fog but also increases the false alarms ratio by a lower magnitude.

**Keywords:** assimilation; wind; COBEL-ISBA; fog; AMDAR; 1D fog forecast

---

## 1. Introduction

Assimilation of wind observations plays an important role in numerical weather prediction (NWP) models to specify the atmospheric dynamics, particularly at the mesoscale. Thus, the initialisation of such models through the assimilation of all available observations is found to be relevant for nowcasting and short-range forecasting of, among others, severe weather events such as fog and heavy rainfall (Strajnar et al., 2015 [1]; De Haan and Stoffelen, 2012 [2]). Bergot and Lestringant (2019) [3] showed that wind at the top of the nocturnal boundary layer plays a significant role during the bifurcation from formation to mature phases of fog layer development. In fact, the authors demonstrate that by modifying the mixing between the nocturnal boundary layer and the residual layer, modestly stronger wind can alter the development of the fog layer and keep fog in a shallow patchy state. As a result of an impact study on fog forecasting, Philip et al. (2016) [4] found that high vertical resolution in a kilometric-scale 3D NWP model leads to stronger nocturnal jet and turbulence at the top of the nocturnal boundary layer. Thus, an improvement of the initial wind field will result in a better forecast of both wind and other meteorological parameters (e.g., temperature and humidity). This will be beneficial for aviation, which highly depends on meteorological information for safety.

However, upper air profile observations are notably lacking (WMO, 2001 [5], 2005 [6]). In fact, radiosonde and aircraft measurements are the main sources of upper air wind (WMO (2016) [7]). The radiosonde are generally launched a few times per day, and do not directly measure wind speed and direction. Through Aircraft Meteorological Data Relay (AMDAR), aircraft-based measurements sample an atmospheric profile in the vicinity of airports and are an excellent means of supplementing upper-air observations obtained by conventional systems, such as radiosondes. However, some companies did not equip their aircraft with AMDAR or they did not activate the system. It should be noted that this information is beneficial for NWP when these observations have been sufficiently quality controlled. Today, the observations from the WMO AMDAR observing system are the output from 12 operational AMDAR national and regional programs in cooperation with some 40 national and international participating airlines. Thus, AMDAR-derived meteorological information supports international air navigation and air traffic management and also supports meteorological applications (e.g., forecasting for aviation, more details are available on the WMO website). Moninger et al. (2003) [8] showed that weather forecasts at both short- and long-term ranges have been improved when AMDAR data are used. In the literature, the AMDAR information has been extensively used in the 3D NWP framework (e.g., Cardinali et al., 2003 [9]; de Haan, 2011 [10]; de Haan and Stoffelen, 2012 [2]) but less in the 1D framework.

Because airport traffic is highly influenced by reduced visibility conditions, a need exists for accurate and updated fog and low-cloud forecasts. For this purpose, single-column models are often used in airports located in flat terrain because they are suitable for the nowcasting of fog events, particularly the radiation fogs (e.g., Bergot and Guédalia 1994 [11], Clark 2002 [12], 2006 [13]; Herzegh et al. 2003 [14]). The National weather service of Morocco (DMN) has implemented a site-specific fog forecasting system at Nouasseur international airport for operational use since 2014. It consists of coupling the one-dimensional model COBEL-ISBA (Code de Brouillard à l'Échelle Locale-Interactions Soil Biosphere Atmosphere, Bergot et al., 2005 [15]) with the high-resolution 3D meso-scale model AROME (Applications de la Recherche à l'Opérationnel à Méso-Echelle, Seity et al., 2011 [16]). Following experiences of other institutes on fog forecasting, local observations have been integrated during the model run. Therefore, additional instruments have been installed close to runways to provide an observational dataset for both model initialization and model diagnostics. This local observation system provides some details on the surface boundary layer state, as well as that of the fog and low-cloud layers. Using the 1D-Var method, the COBEL-ISBA model assimilates the information from this local observation system to produce initial profiles of temperature and specific humidity. In fact, initial conditions have a great impact on the skill of the forecast (Roquelaure and Bergot 2007 [17]; Rémy and Bergot 2009 [18]). It is the first time that wind profiles are assimilated in the framework of a single-column model, particularly into a fog/low clouds forecasting system.

The objective of this study is to assess the impact of wind assimilation on the forecast and data assimilation systems into the single-column model, COBEL-ISBA. Three sources of wind data are used: SYNOP, AMDAR, and 3D mesoscale NWP data. The next section gives a description of the single-column model COBEL-ISBA and its assimilation system, followed by a description of wind data used in the assimilation experiments in Section 3. The experimental design is detailed in Section 4. Then, Section 5 is devoted to the results of the experiments and a discussion of the impacts on temperature and specific humidity analysis and also on the forecasting of these parameters in the upper levels and of the near-surface thermodynamical parameters. The impact on fog forecasting is also discussed in this section. The final section is devoted to the conclusion and recommendations.

## 2. Data Assimilation and Fog Forecast Systems in COBEL-ISBA

### 2.1. The COBEL-ISBA Fog Forecast System

COBEL-ISBA is a high-resolution single column numerical model originally designed to simulate the evolution of the stable atmospheric boundary layer vertical structure at the local scale. It was developed at the Laboratoire d'Aérologie, Paul Sabatier University in collaboration with Meteo-France and Quebec University at Montreal, Canada. The high-resolution vertical grid of the model is logarithmic and consists of 30 levels (0.5, 1.6, 3.0, 4.7, 6.8, 9.2, 12.2, 15.9, 20.2, 25.5, 31.9, 39.6, 49.0, 60.3, 74.0, 90.5, 110.5, 134.7, 164.0, 199.4, 242.2, 293.9, 356.5, 432.3, 523.8, 634.6, 768.6, 930.6, 1126.5, 1363.5 m) between 0.5 and 1360 m, with 20 levels below 200 m. It is coupled with the multilayer surface-vegetation-atmosphere transfer scheme ISBA-DF (Boone et al. 1999 [19], 2000 [20]). The latter runs with seven soil levels, from 1 mm to 1.7 m below the surface.

The COBEL-ISBA model also integrates external mesoscale forcings from 3D mesoscale NWP models (e.g., AROME at DMN; WRF in Greece, Stolaki et al. (2012) [21]) to take into account the influence of possible horizontal heterogeneities on the local forecast scale. These forcings include horizontal advection of potential temperature and moisture, geostrophic wind, local pressure tendencies, and cloud cover. The forcings computation requires the following NWP data at 15 levels (20, 50, 100, 250, 500, 750, 1000, 1250, 1500, 2000, 2500, 3000, 4000, 5000 and 6000 m heights above ground): temperature, specific humidity, potential temperature, the two horizontal wind components, and pressure, as well as downward short-wave and long-wave radiative fluxes under clear skies at the ground. This model incorporates more sophisticated parametrizations including a detailed radiation transfer scheme (Vehil et al., 1989 [22] for the longwave part; Fouquart and Bonnel (1980) [23] for the shortwave part), a parametrization of the boundary layer turbulent mixing under stable (Estournel and Guedalia, 1987 [24]), neutral (Delage, 1974 [25]) and unstable (Bougeault and Lacarrère, 1989 [26]) conditions, and a microphysical parametrization adapted to fog and low clouds (For more details, see Bergot et al. (2005) [15]). Note, that fog from COBEL-ISBA is defined based on visibility at the second level (1.65 m) when it is below 1 km. Besides, horizontal visibility is diagnosed using only liquid water content (LWC) following Kunkel (1984) [27].

### 2.2. The COBEL-ISBA Data Assimilation System

The COBEL-ISBA data assimilation system is presented in detail in Bergot et al. (2005) [15]. It generates the initial conditions by a two-step assimilation scheme. First, this system generates a best linear unbiased estimator (BLUE) $x^a$ for initial conditions using information from a dedicated local observation system $y^o$, a first guess $x^b$ (i.e., a previous 1-h COBEL-ISBA forecast) and profiles from the mesoscale AROME NWP model:

$$x^a = x^b + K(y^o - Hx^b) \tag{1}$$

where

$$K = BH^T(HBH^T + R)^{-1} \tag{2}$$

$B$ and $R$ are the error variance and covariance matrices of the background and the observations, and $H$ is the forward operator that interpolates information from the model grid to the observation grid. $K$ is the Kalman gain that accomplishes the observations weighting. The local observation system gives information about the state of the atmosphere below 20 m and the maximum height of the COBEL-ISBA domain is 1360 m, it is necessary to have additional data. Thus, AROME data are taken as pseudo-observations for the upper levels of the model domain, so a part of $R$ corresponds to the error variances and covariances of the AROME profiles.

In the operational setup, the assimilation scheme is multivariate where background error statistics are flow-dependent and they are computed using the ensemble Monte Carlo method. In fact,

an ensemble of N members (in this study $N = 12$) is used to estimate $B$ in a flow-dependent way by taking covariance statistics of the differences between each member $x^b$ and the ensemble mean $\langle x^b \rangle$:

$$B \approx \langle (x^b - \langle x^b \rangle)(x^b - \langle x^b \rangle)^T \rangle \tag{3}$$

where the members represent previous COBE-ISBA forecasts $x^b$ valid at the same time $t_0$ and issued from runs at $t_0 - 1h$, $t_0 - 2h$, $t_0 - 3h$, ..., $t_0 - 12h$. The ensemble mean $\langle x^b \rangle$ is computed based on the N members at $t_0$ for an estimate of the instantaneous $B$ matrix.

At the initialization time of COBEL-ISBA, the local observations system allows the fog and low cloud detection due to visibility and ceiling measurements. When this occurs, an additional step in the assimilation scheme is performed to modify the generated initial profiles from the first step. However, the thickness of the cloud layer is not measured directly. To overcome this deficiency, Bergot et al. (2005) [15] developed a minimization algorithm using radiative flux observations at 2 and 20 m to estimate this thickness. The best estimate of the initial fog thickness is the one that minimizes the error between modeled and observed radiative fluxes. The relative humidity profile is then modified within the saturated layer. The soil temperature and water content profiles used to initialize ISBA are obtained directly by interpolation of soil measurements.

### 2.3. Observation Screening and Thinning

As described by Andersson and Jarvinen (1999) [28], background quality control is performed for all data that are intended to be used during the assimilation process. Thus, data are considered as suspect if the departure from the background $d = y^o - Hx^b$ exceeds a multiple $\beta$ of its expected error. In fact, the variance of the departure background can be estimated by $\sigma_b^2 + \sigma_o^2$, assuming that observation ($\sigma_o$) and background ($\sigma_b$) errors are uncorrelated. Therefore, rejection occurs if $d > \beta\sqrt{\sigma_b^2 + \sigma_o^2}$ (Andersson et al. 2000 [29]). For aircraft wind data, the value of $\beta$ is set to 5, as in Cardinali et al. (2003) [9].

Aircraft data are checked for redundancy. If two reports have the same metadata (time, latitude, longitude, and flight level) and the measured values are identical, then the measurements are considered redundant and only one is used. Besides, a certain thinning distance is applied during observation screening. In our study, thinning is performed by selecting one report per flight within a distance of 30 km around the airport. In addition, aircraft data are checked for the vertical consistency and the duplicated levels are removed.

Next, aircraft-based wind data are scanned for whitelisting. This approach is applied in the assimilation experiments and consists of aircraft selection with reliable observations. This is beneficial for real-time applications because it prevents a new or reregistered aircraft with unknown error characteristics to enter the data assimilation process without a priori check. Note, that no special procedure for bias correction has been done during this preliminary study.

## 3. Observations

### 3.1. Surface and Mast Observations

The local observation system used at Nouasseur airport is designed to provide up-to-date information on the lower layers of the atmosphere. In fact, for radiation fog, the layers of the atmosphere in contact with the ground those are the most important for the prediction of this fog type. At Nouasseur airport, the local observation system consists of the following:

- A measurement mast that provides observations of temperature and humidity at 1, 5, 10 and 20 m, as well as short- and long-wave radiative fluxes at 2 and 20 m.
- A weather synoptic station that provides 2 m temperature and humidity, as well as visibility, 10 m wind speed and direction, pressure and ceiling.
- Soil temperature and water content in the ground at $-5$, $-10$, $-20$, $-50$ and $-100$ cm.

### 3.2. Aircraft Observations

Wind data are derived from aircraft measurements (speed of the aircraft and its position). In addition, aircraft sensors also measure ambient temperature and pressure. Then, an atmospheric profile can be generated when measurements are taken during takeoff and landing (see WMO (2003) [30] and de Haan (2011) [10] for more details). According to Benjamin et al. 1991 [31] and WMO 1996 [5], the aircraft provide automated reports of wind measurements with an accuracy of 1–2 m·s$^{-1}$ for vector wind.

Taking into account the high spatial and temporal variability of the wind, in particular, in the atmospheric boundary layer, the wind profiles are extracted from AMDAR's messages within a radius of 30 km around the Nouasseur airport for heights not exceeding 1360 m above ground level (top of the model COBEL-ISBA) and in a window of 20 min [HH − 10 min, HH + 10 min]. Regarding the temporal distribution of vertical wind profiles during the day, we presented in Figure 1a the hourly frequency of AMDARs retained after quality control and thinning over the winter period (December 2015–February 2016). It is seen clearly that high frequencies are observed at 14 UTC, 08 UTC, and 18 UTC, respectively. In fact, there are very few or no observations during the night during the study winter period, particularly between midnight and 06 UTC. On the other hand, statistics of measurements availability between two successive levels of the COBEL-ISBA model are plotted in Figure 1b, which highlights the high frequency of the wind components for the upper levels of the COBEL-ISBA model with a clear absence of measurements at the lower levels of the atmospheric boundary layer (especially below 60 m).

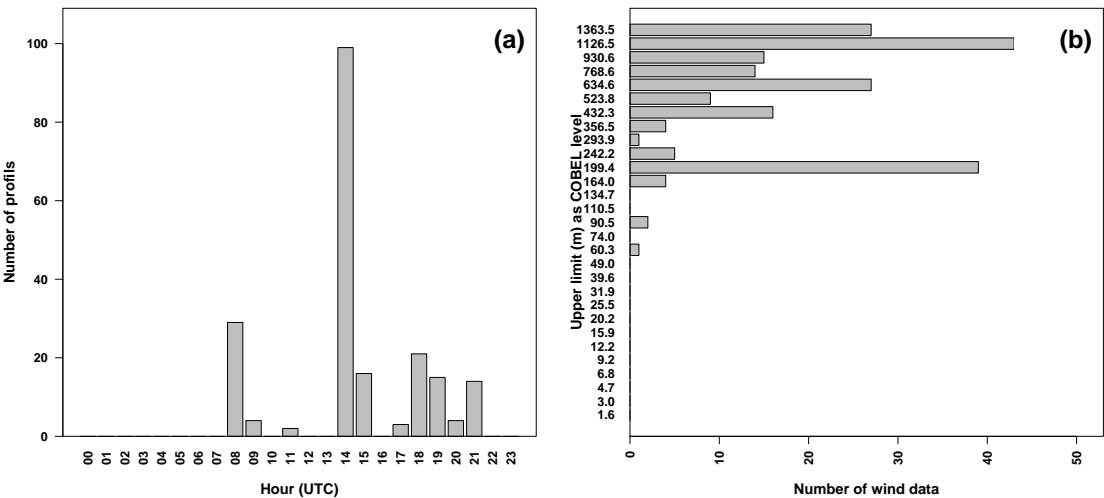

**Figure 1.** Statistics of wind AMDAR frequency within a radius of 30 km around Nouasseur international airport and below 1360 m (top of the COBEL-ISBA model) over the winter period (December 2015–February 2016): (**a**) Hourly frequency, and (**b**) Vertical frequency as function of COBEL-ISBA levels.

## 4. Experiments Design

As in operational use, COBEL-ISBA is run at 1 h intervals and provides up to 12 h of fog forecasts. Then, an hourly rapid updating cycle (RUC) will be applied to assimilate the AMDAR wind observations, along with SYNOP and NWP wind data. Although all experiments are performed off-line, all the settings are kept as close as possible to operational practice to have an assessment of the impact of these observations representative for operational practice.

Concerning the initialization of the wind profile in the current operational configuration of COBEL-ISBA, the 10 m wind from SYNOP messages, issued from the synoptic weather station, is the only available measurement. Thus, to reconstruct the wind profile, the geostrophic wind profile from the NWP model is used following Stull (1988) [32]. This method has been successfully

used in previous studies (for more detail see Bergot (1993) [33], Guedalia and Bergot (1994) [11], or Roquelaure (2004) [34]).

At the initialization time of the model COBEL-ISBA, wind data can be extracted from the SYNOP (observation at 10 m), AMDAR or NWP model forecasts (AROME model is this study with 2.5 km as horizontal resolution and 90 vertical levels) considered as pseudo-observation to fill the absence of observations beyond 10 m. It should be noted that for the AROME levels, only those below 1360 m (top of the COBEL-ISBA model) are retained. Indeed, NWP data were extracted at the following levels: 20, 50, 100, 250, 500, 750, 1000, 1250 m. The assessment of the wind impact on the multi-variate assimilation system of COBEL-ISBA is carried out through data denial tests, which were conducted for a winter period (December 2015–February 2016) and are summarized in Table 1. A data denial test compares results from a control simulation analysis and forecasts using all data sources with simulations (i.e., reruns) in which different sources of data were removed individually. Differences in forecast errors between the control run and various tests are then used as a measure of the impact of each data source. In the tests discussed here, only AMDAR and NWP wind observations were denied. The control simulation (DA-Wind) is carried out with assimilation of all available sources of wind observations. To assess the overall impact of wind assimilation, a second experiment (DA-NoWind) is performed without assimilation of the wind, as is used currently in operation. To highlight the contribution of the assimilation of the wind data from NWP model and AMDAR, two other experiments (DA-Wind-NoNWP and DA-Wind-NoAMDAR) are carried out without assimilation of each type of data separately.

**Table 1.** Data assimilation experiments.

| Experiment | Details |
| --- | --- |
| DA-NoWind | No wind assimilation, only 3D NWP geostrophic wind is used in initial conditions |
| DA-Wind | Assimilation of all sources of wind information (3D NWP, AMDAR and SYNOP) |
| DA-Wind-NoNWP | 3D NWP wind data are not assimilated |
| DA-Wind-NoAMDAR | AMDAR wind data are not assimilated |

## 5. Results

### 5.1. Background Errors Diagnosis

Based on statistical estimation theory, data assimilation uses the available observations and an initial guess or background (i.e., a short-term forecast from the previous assimilation-forecast cycle) to find the best possible initial state of the atmosphere, called analysis. From this initial state, a forecast can be computed through integration in time. Both sources of information are prone to errors (model error and observations inaccuracies). Thus, the data assimilation system accounts for this through the observation and background error covariance statistics ($R$ and $B$ matrix).

In the framework of the multivariate mode, the assimilation system into COBEL-ISBA estimates the flow-dependent background covariances for each run of the model and takes the cross-correlations between temperature, humidity and wind components into account. Thus, to highlight the main differences between the $B$ matrix for the experiments with wind assimilation and those that will be used later, we will focus on the background error standard deviation of the control parameters: temperature, specific humidity, zonal and meridional components of wind. The vertical distributions of the standard deviation for these parameters, averaged over the winter period, are plotted in Figure 2. In fact, the $B$-matrix spreads out information in the vertical direction and allows background information to be weighted against observational information according to their respective uncertainties. It should be noted that a $B$-matrix with smaller standard deviation for a given parameter implies that the background will be highly weighted during the assimilation process. Thus, the background will be more trusted in this case.

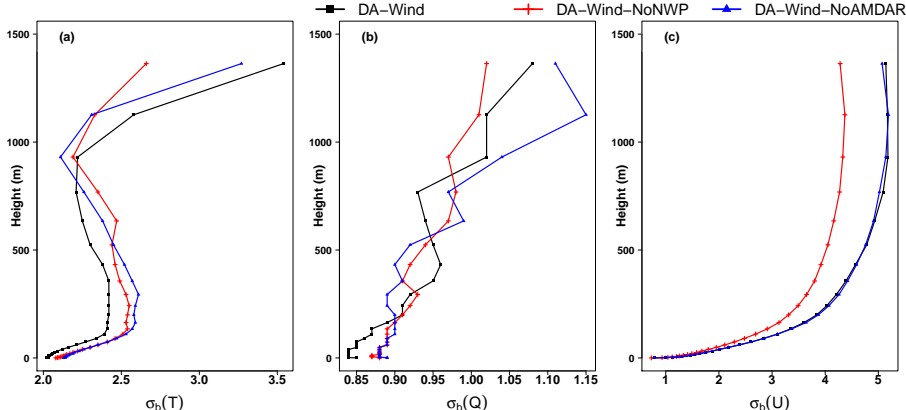

**Figure 2.** Background error standard deviation for (**a**) temperature (°C), (**b**) specific humidity (g·kg$^{-1}$), and (**c**) zonal wind (m·s$^{-1}$) for DA-Wind, DA-WindNoNWP and DA-WindNoAMDAR experiments.

For temperature (Figure 2a), we notice that the *B*-matrix associated with DA-Wind experiment (all wind observations are assimilated) has the smaller standard deviation in the lower levels of the atmosphere, while the DA-Wind-NoAMDAR configuration shows the higher values of standard deviation. In the upper levels, when NWP wind data are not assimilated, the smaller values of standard deviation are associated with the DA-Wind-NoNWP experiment.

For specific humidity (Figure 2b), it is seen clearly that the background has similar weights for the three configurations, with a slight difference in the upper levels where the *B*-matrix associated to the DA-Wind-NoNWP Experiment (SYNOP and AMDAR wind data are assimilated) has a smaller standard deviation.

For the wind components (Figure 2c for zonal wind and Figure S1 in the supplement for meridional wind), it is found that standard deviation increases as the information is spread out in the vertical direction for all configurations. Although, the configuration where SYNOP and AMDAR data are assimilated shows the lowest values of the standard deviation in the middle and upper levels.

Figure 3 illustrates the cross-correlations between zonal wind component (U) with temperature T (Figure 3a) and specific humidity Q (Figure 3b), averaged over the December 2015–February 2016 period and issued from DA-Wind experiment. It is seen clearly from this figure that the *B*-matrix spreads information to other variables and imposes balance by permitting multivariate error covariance. Thus, we notice that cross-correlations are stronger in the upper levels for zonal wind and lower levels for temperature. The same finding is noticed for specific humidity with lower correlations. Similar results have been found for the cross-correlation of these parameters (T and Q) with the meridional wind component (V).

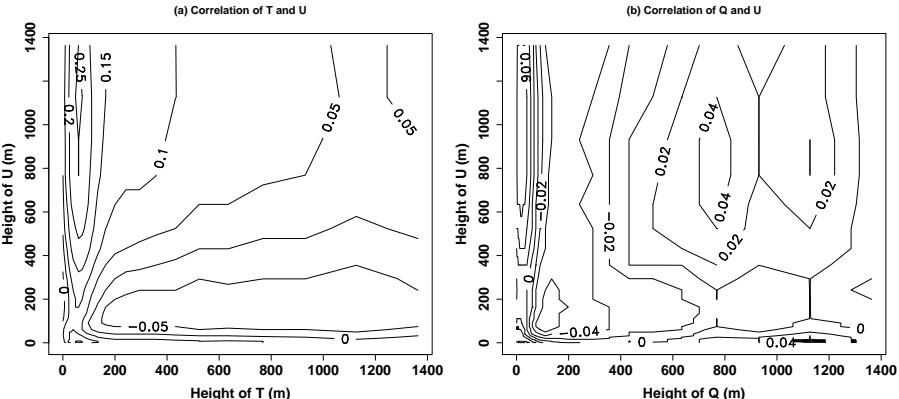

**Figure 3.** Correlations of zonal wind component U with: (**a**) temperature T, and (**b**) specific humidity Q averaged over the winter period (December 2015–February 2016).

Overall, this analysis showed that the background is highly weighted in the upper levels of the COBEL-ISBA model when AMDAR data are assimilated without NWP data. Although, previous research has shown that many factors influence the background error covariance, such as the studied geographical area, and the used model resolution (Brousseau et al., 2011 [35]), the period used for B matrix computation (Monteiro and Berre, 2010 [36]; Berre et al., 2013 [37]), and also, weather regimes (Brousseau et al., 2012 [38]). These aspects have not been studied in the present preliminary study. In the following section, the impact of wind assimilation on the vertical profiles of temperature and specific humidity background is assessed.

## 5.2. Impacts on Temperature and Specific Humidity Background

In the framework of multivariate data assimilation, the cross-correlations of the two components of wind, temperature and specific humidity in the background were taken into account. To assess the impact of these cross-correlations on the quality of initial conditions, the vertical profiles of Observation minus Background (OmG) statistics (Bias and root-mean square error—RMSE), averaged for all the runs over the winter period, for temperature are plotted in Figure 4 (for specific humidity see Figure S2 in the Supplementary Materials) for the four experiments (See Table 1). Improvement is indicated by proximity to the dashed zero line. The Bias or mean error is given by

$$Bias = \frac{1}{n} \sum_{i=1}^{n} (f_i - O_i) \tag{4}$$

where $f_i$ is the forecasted value, $O_i$ the observed value and $n$ is the total number of observations.

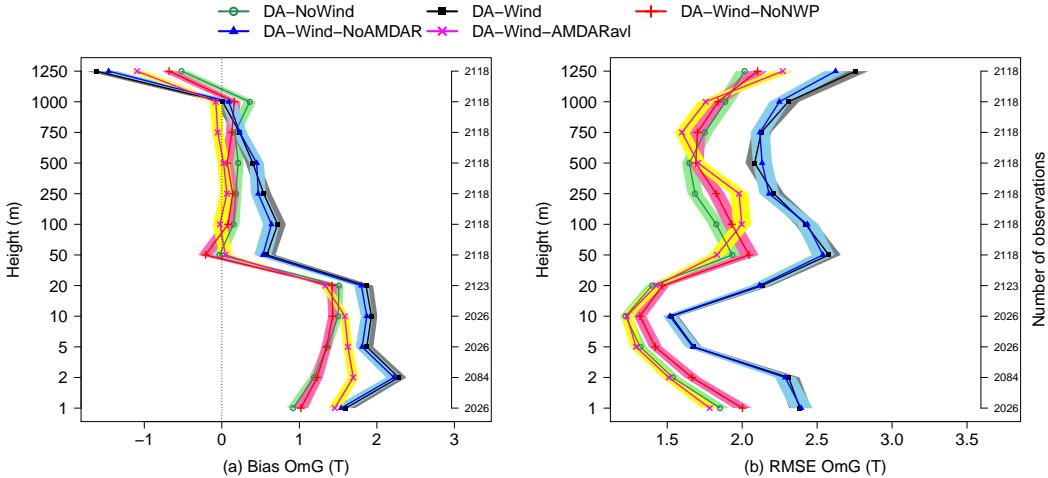

**Figure 4.** Background departure (OmG) statistics for temperature (T), averaged over the winter period (December 2015–February 2016): (**a**) Bias, and (**b**) RMSE. The number of observations used is shown on the right. The shaded areas represent the confidence intervals at 95%.

The RMSE is given by

$$RMSE = \sqrt{\frac{1}{n} \sum_{i=1}^{n} (f_i - O_i)^2} \tag{5}$$

The observed state for temperature and specific humidity contains the 2 m observations from the synoptic station, as well as measurements issued from the mast (at 1, 5, 10 and 20 m). To fill the gap of observations in the upper levels, the observed state also contains NWP data at 20, 50, 100, 250, 500, 750, 1000, and 1250 m.

For both statistics, confidence intervals are plotted in order to determine whether the impacts are statistically significant or not. Indeed, confidence intervals give information about how much variability

there is, and allow the reader to compare the magnitude of a difference between experiments. Thus, a wider confidence interval has considerable variability. In addition, when comparing the same statistic between two different experiments, if confidence intervals overlap by quite a lot, they definitely are not statistically different.

The first finding from Figure 4 and Figure S2 is that assimilating all sources of wind data (NWP, SYNOP, and AMDAR) has a negative impact on OmG Bias and RMSE for the temperature at all levels and on OmG RMSE for the specific humidity at the upper levels by comparing DA-Wind and DA-NoWind experiments. To identify which wind data source is responsible for that, NWP and AMDAR data were removed individually. Therefore, it is found that when AMDAR is removed, the negative impact remains (DA-Wind and DA-Wind-NoAMDAR are very similar), indicating that NWP data has a detrimental impact. Indeed, the statistics are better when NWP wind data are removed (DA-Wind-NoNWP).

Compared to DA-NoWind, the behavior of DA-Wind-NoNWP statistics is similar with a clear improvement in temperature Bias and specific humidity RMS error at all levels, while a slightly positive impact is observed in the upper levels for temperature RMS error and specific humidity Bias. As stated before in the methodology section, all experiments have been designed to be similar to operational practice. Then, when AMDAR data are missing for a given run, wind assimilation is not performed. To emphasize the benefit from assimilating AMDAR with SYNOP, statistics of OmG for temperature and specific humidity issued from DA-Wind-NoNWP, only for runs when AMDAR data are available (Figure 1), were computed and called DA-Wind-AMDARavl (Figure 4). It is found that DA-Wind-AMDARavl for temperature has the best RMSE, with a slight degradation between 100 m and 250 m, and the best bias above 20 m. For specific humidity, DA-Wind-AMDARavl has good Bias and RMSE in lower levels of the atmosphere below 100 m. Overall, the superior fit of observations to the background fields, at analysis time, when assimilating AMDAR without NWP wind data, provides further evidence of the importance and quality of the AMDAR data. The beneficial impact on the background fields is propagated in forecasts. This will be assessed in the following section.

*5.3. Impacts on Forecast*

One goal of this research is to assess the impact of wind data assimilation, particularly issued from AMDARs, on forecasts with a focus on future operational applications. Thus, the operational quality forecast from an hourly rapid update cycle will be compared to many NWP configurations with wind observations. First, comparing DA-NoWind (current operational configuration) and DA-Wind (configuration when all wind sources are assimilated) experiments will highlight the added value of wind assimilation and its impact on the forecasting system. Besides, comparing DA-Wind, DA-Wind-NoNWP and DA-Wind-NoAMDAR experiments will point out the influence and the contribution of AMDAR and NWP wind data separately to the overall impact of wind assimilation. In the following sections, this impact will be assessed on the forecasting of temperature and specific humidity at the upper levels of the COBEL-ISBA fog forecasting system, and also on the forecasting of the conventional near-surface meteorological parameters (2 m temperature, 2 m relative humidity, and 10 m wind speed). In addition, the impact on the fog forecasting will be evaluated.

5.3.1. Impacts on Forecasted Upper-Level Temperature and Specific Humidity

To assess the quality of the forecast for continuous parameters such as temperature and specific humidity, we use the mean error (Bias) and the root-mean-square error (RMSE). In Figure 5, the statistics (Bias and RMSE) of the model forecasts for the four experiments (DA-NoWind, DA-Wind, DA-Wind-NoNWP and DA-Wind-NoAMDAR) are shown at different vertical levels (50 m, 500 m and 1000 m) for temperature (Figure 5a,b) and specific humidity (see Figure S3 in the Supplementary Materials).

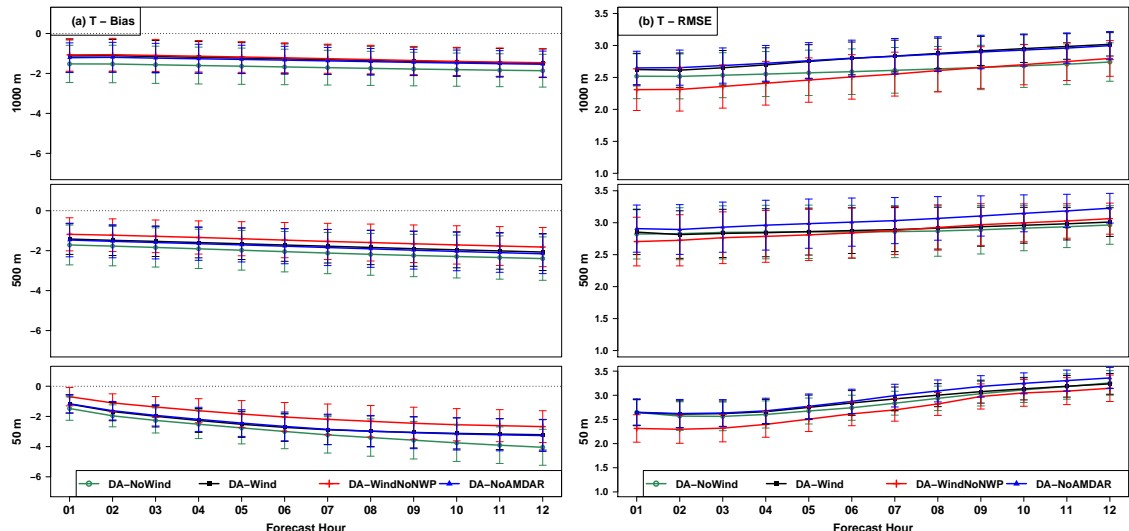

**Figure 5.** Time series of statistics as function of forecast hours (up to 12 h) for temperature at the following vertical levels: 50 m, 500 m and 1000 m: (**a**) Bias (left), and (**b**) RMSE (right). The error bars represent the confidence intervals at 95%.

For temperature, the first observation from this figure is that the Bias and RMSE during the forecasts for the levels shown are reduced when SYNOP and AMDAR data are used (DA-Wind-NoNWP experiment). It is also seen clearly that DA-Wind and DA-Wind-NoAMDAR experiments fit well at all levels for Bias, indicating the high impact of NWP wind data when all wind observations are assimilated. This is mainly due to the high availability in time of these data in comparison with that of AMDAR's data. The latter is only available in the upper levels (especially above 500 m) and for some specific hours during the day (Figure 1).

At upper levels, a positive impact of assimilating AMDAR and SYNOP wind data, without NWP data is observed; however, this impact disappears after a certain forecast length, and its duration differs for the shown levels. Regarding the benefit in RMS error for temperature, it is found that it is lost during the first forecast hours, especially at 50 m when comparing DA-NoWind and DA-Wind-NoAMDAR and DA-Wind (Figure 4b vs. Figure 5b). This would mean that observation weights are reasonable, but the spread of information is erroneous. This could be due to the structure of the background error covariance, which is instantaneous and constructed based on the 12 previous forecasts. Another possible explanation is the fact that not all observation data used in the comparison are usually assimilated in the hourly runs due to their rejection during the quality control. However, more investigation is needed to clarify this discrepancy further.

On the other hand, it is found that the statistics distribution (Bias and RMSE), issued from the limited AMDAR interval (few hours during the day), depends on the runs. In fact, Figure S4 (in the Supplementary Materials) represents the time series of bias and RMSE for temperature for runs where AMDAR data are available, as a function of forecast hours (up to 12 h) at the following vertical levels: 50 m, 500 m and 1000 m. It is seen clearly that the daily cycle of temperature, for example, is observed in the evolution of the statistics, particularly in the lower levels of the atmosphere.

For specific humidity, a slight degradation is found in Bias with a mixed impact on the vertical levels (Figure S3b). Indeed, it is found that DA-Wind outperforms the other configurations in the middle and upper levels, while DA-NoWind has a better Bias in the lower levels of the atmospheric boundary layer. Besides, we notice an improvement in RMS error at all levels, over the whole forecast range, when all wind data are assimilated (DA-Wind). From Figure S3b, the benefit of assimilating AMDAR data is most visible and its impact is clear in reducing the RMS error for specific humidity at all levels.

5.3.2. Impacts on Forecasted Near-Surface Thermodynamical Parameters

We present here the results obtained by the different experiments in forecasting the near-surface thermodynamical parameters at Nouasseur airport using COBEL-ISBA. The current operational configuration (DA-NoWind) is considered as a baseline for comparison, thus we compare the results in terms of forecast accuracy scores (root-mean-square error (RMSE) and mean absolute error (MAE)), for which a skill score is defined as

$$SS_{score} = 1 - \frac{score_{exp}}{score_{ctl}} \tag{6}$$

where *score* can be MAE or RMSE, $score_{exp}$ is the score for the configuration *exp* and $score_{ctl}$ its equivalent for the reference configuration. Note that the closer $SS_{score}$ is to zero, the more similar the experiment is to the reference. On the other hand, positive values of $SS_{score}$ imply that the experiment outperforms the reference, whereas negative values of $SS_{score}$ imply the reference is a better configuration for prediction than the experiment.

We first consider the importance of assimilating wind and its impact on the COBEL-ISBA forecasting system near the surface, by comparing the configurations with and without wind assimilation. In Figure 6, we have plotted the skill scores of MAE and RMSE for the 2 m temperature, 2 m relative humidity and 10 m wind speed issued from DA-Wind, DA-Wind-NoNWP and DA-Wind-NoAMDAR compared to DA-NoWind. Regarding the DA-Wind experiment, this figure points out that wind assimilation improves the two scores (MAE and RMSE), except for in the first forecast hour. To identify which source of data positively impact the performance of the model when NWP and AMDAR data are assimilated separately, we compare the skill scores of DA-Wind-NoNWP and DA-Wind-NoAMDAR. In all cases, we observe that DA-Wind-NoNWP (SYNOP and AMDAR data are assimilated) outperforms the two other configurations and improves the MAE and RMSE for all thermodynamical parameters over the whole forecast range.

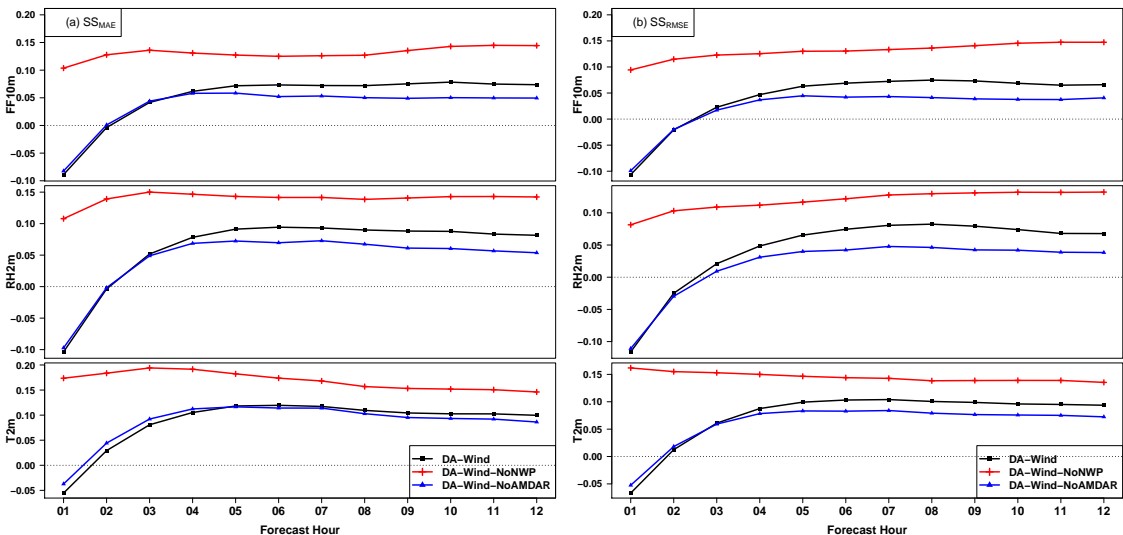

**Figure 6.** Evolution of skill scores of : (**a**) MAE (left) and (**b**) RMSE (right), as function of forecast hours, for near-surface (at 2 m) temperature and relative humidity, and 10 m wind speed.

5.3.3. Impacts on Fog Forecasting

To assess the impact of wind assimilation on the fog forecast system into the one-dimensional COBEL-ISBA model, the verification scores for categorical forecasts, such as the probability of detection (*POD*) and the false alarm ratio (*FAR*) for accuracy are used in this study. *POD* is the fraction of observed events that were correctly predicted to exist, and *FAR* is the fraction of predicted events that are not observed:

$$POD = \frac{a}{a + c} \tag{7}$$

$$FAR = \frac{b}{a+b} \tag{8}$$

where $a$ is the number of observed and forecasted fog events, $b$ is the number of not observed and forecasted fog events, and $c$ is the number of observed and not forecasted fog events.

These scores were computed for the four experiments detailed in Table 1 as a function of forecast time. During the result's analysis, we will focus on the skill score, which measures the forecast accuracy with respect to the accuracy of the operational forecast (DA-NoWind configuration in this study) as a benchmark. This will assess the impact of each configuration. Thus, the skill score is defined as the difference between the score for a given configuration ($score_{exp}$) and the operational one's score ($score_{ctl}$), normalized by the score obtained for a perfect forecast ($score_{perf}$) minus the operational forecast score (for perfect forecasts POD = 1 and FAR = 0).

$$SS_{score} = \frac{score_{exp} - score_{ctl}}{score_{perf} - score_{ctl}} \tag{9}$$

It should be noted that positive values for skill score of POD and FAR imply that the experiment outperforms the reference, while both of them are similar when the skill score is closer to zero.

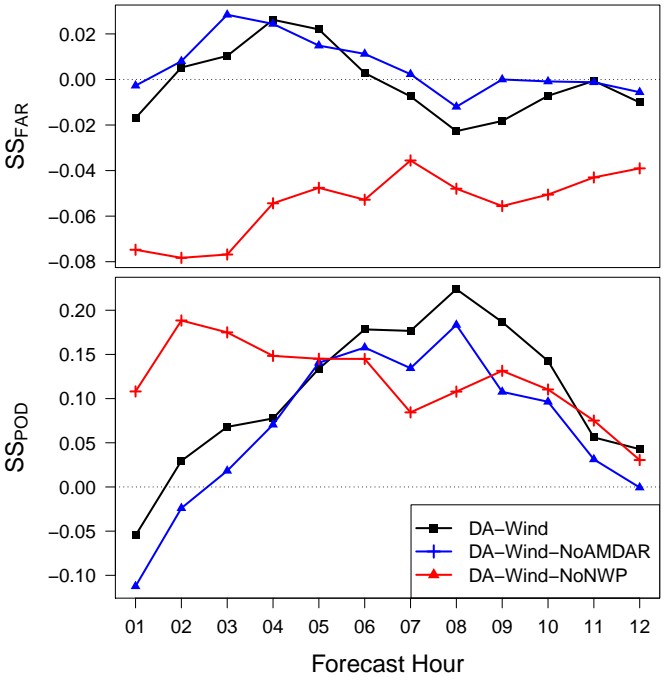

**Figure 7.** Evolution of skill scores of POD and FAR as a function of forecast hours.

To compare the various forecasts, the evolution of the skill scores of POD and FAR are plotted, in Figure 7, as a function of the forecast time for forecasts issued from the configurations with wind assimilation against DA-NoWind as a benchmark. Regarding the POD, this figure shows that the COBEL-ISBA forecast issued from DA-Wind-NoNWP is slightly better than the DA-NoWind forecast (current operational configuration) at all forecast times ($SS_{POD}$ varies between 0.05 and 0.2), which is associated with a slight degradation of FAR ($SS_{FAR}$ ranges between $-0.04$ and $-0.08$). On the other hand, when NWP wind data are assimilated, both DA-Wind and DA-Wind-NoAMDAR have similar skills in comparison with the benchmark. In fact, these configurations are similar to the current operational configuration during the first forecast hours and then, POD is improved associated with the same magnitude of FAR. These results demonstrate that assimilation of SYNOP and AMDAR wind measurements slightly improves the probability of detection of fog but also increases the false alarms ratio with lower magnitude. In fact, even with dedicated observations, uncertainties remain in

both initial conditions and mesoscale forcings under the hypothesis of a perfect model. This, in turn, impacts the fog forecasting system in COBEL-ISBA (Roquelaure and Bergot, 2007 [17]).

## 6. Conclusions and Discussion

As poor visibility conditions have a considerable influence on airport traffic, a need exists for accurate and updated fog and low cloud forecasts. For this purpose, the COBEL-ISBA local numerical forecast system has been implemented at Nouasseur international airport, Casablanca, Morocco. This fog/low-clouds forecast system assimilates (using the 1D-Var method) the information from a local observation system designed to provide details on the state of the surface boundary layer, as well as that of the fog and low-cloud layers, to produce initial profiles of temperature and specific humidity. At the initialization time of the model COBEL-ISBA, wind observations can be extracted from the SYNOP (observation at 10 m), AMDAR or a 3D NWP AROME forecasts considered as pseudo-observations to fill the absence of observations beyond 10 m. The impact of wind on the assimilation and forecast systems in COBEL-ISBA is assessed in this preliminary study.

The impact is assessed by performing NWP experiments with data denial tests in the COBEL-ISBA model, configured to be close to the settings used operationally at the Nouasseur airport. Thus, an hourly assimilation cycle is applied and the reported impacts are representative of the operational practice. Results reveal that assimilation of SYNOP and AMDAR wind observations can substantially improve vertical profiles of atmospheric variables (temperature and specific humidity) in the background, increasing their similarity to the observations, particularly in the upper levels.

The beneficial impact on analyses is propagated into the forecasts. Thus, a positive impact is found on temperature forecasts at the upper levels when AMDAR data are assimilated without NWP data. The latter has shown a detrimental impact in all experiments. For specific humidity, a slight degradation is found in Bias with a mixed impact on the vertical levels associated with an improvement in RMS error at all levels, over the whole forecast range. Near the surface, a slight but consistent thermodynamical parameters forecast improvement is found up to the end of the 12-h forecast range. Besides, assimilation of SYNOP and AMDAR wind measurements improves slightly the probability of detection of fog but also increases the false alarms ratio with lower magnitude.

The studied airport is located in a fog-prone coastal area (Bari, 2015 [39]) where advection–radiation fog events are the most common fog type. The analysis of local meteorological and synoptic conditions over this region shows that advective processes at mesoscales associated with sea-breeze circulation during the afternoon and followed by radiative processes early in the night (local cooling) often lead to fog formation over this region (Bari et al. 2015 [40]). Similar wind impact on fog formation has been found by Rayznar (1977) [41]. Other research studies (e.g., Cuxart and Jimenez, 2012 [42]) have shown that fog could be affected by the local wind system, particularly for slope winds over complex terrains. Indeed, Cuxart and Jimenez (2012) [42] found that the mountain slopes induces the generation of local flow above the fog layer, which interacts with the top of the fog layer. Prtenjak et al. (2018) [43] focused in their study on the analysis of thermodynamic conditions associated with the formation and dissipation of fog and its relationship with katabatic flow. The authors pointed out that local flow aloft along the slope of Medvednica is responsible for bringing warmer air over the city and, consequently, strengthen and maintain the temperature inversion over the city center. In this preliminary systematic study, the impact of wind assimilation on thermodynamic parameters and also, of fog forecasting, are shown. Although, this impact needs to be investigated further through future studies of several fog event cases. In fact, the studied region has a complex landscape with significant variations in land surface characteristics (urban, suburban, and rural areas), along with high levels of pollution and low mountains (height below 1000 m). All of these factors provide a wide range of influences that could potentially affect the dynamic behavior and microphysical characteristics of fog and low clouds.

In addition, this preliminary impact study provides evidence of the importance and quality of AMDAR wind data in a dedicated local assimilation system, even if they are irregularly distributed

in space and time. This is a clear indication that the availability of more aircraft measurements would further improve the analyses and forecasts. Thus, every effort should be made in equipping inter-continental and regional aircraft with AMDAR capabilities. In addition, it is expected that the assimilation of temperature from AMDAR will be beneficial for both assimilation and forecast systems in COBEL-ISBA. This will be elaborated in further study. On the other hand, accurate visibility forecasts are beneficial for air-traffic managers to optimize air-traffic control at international airports. In this preliminary study, it is found that the fog forecast remains a challenge due to uncertainties in initial conditions even with the dedicated local observing system. The next step is the evaluation of the impact of all sources uncertainties for COBEL-ISBA forecasts at Nouasseur airport, within the perspective of developing a Local Ensemble Prediction System for operational use.

**Supplementary Materials:** The following are available online at http://www.mdpi.com/2073-4433/10/10/615/s1, Figure S1: Background error standard deviation for meridional wind (m·s$^{-1}$) for DA-Wind, DA-WindNoNWP and DA-WindNoAMDAR experiments. Figure S2: Background departure (OmG) statistics (Bias, left; RMSE, right) for specific humidity, averaged over the winter period (December 2015–February 2016). The number of observations used is shown on the right. Figure S3: Time series of statistics (Bias, left; RMSE, right) as a function of forecast hours (up to 12 h) for specific humidity at the following vertical levels : 50 m, 500 m and 1000 m. Figure S4: Time series of statistics (Bias, left; RMSE, right), for runs where AMDAR data are available, as a function of forecast hours (up to 12 h) for temperature at the following vertical levels : 50 m, 500 m and 1000 m. The runs of day are in red and those of the night are in blue.

**Funding:** This research received no external funding.

**Acknowledgments:** The author would like to thank Thierry Bergot (CNRM/Meteo-France) for helpful comments on the first version of this article. My colleagues (Sbii, Sahlaoui and Hdidou) from the CNRMSI/SMN team, at the Moroccan Weather Service (DMN), are also gratefully acknowledged for discussions around this research. The author thank the staff of the Nouasseur meteorological station and Noura Fadaili from CNRMSI/SMN for their daily monitoring and maintenance of the local observation system implemented in the airport.

**Conflicts of Interest:** The author declares no conflict of interest.

## Abbreviations

The following abbreviations are used in this manuscript:

| | |
|---|---|
| AMDAR | Aircraft Meteorological Data Relay |
| NWP | Numerical Weather Prediction |
| COBEL-ISBA | Code de Brouillard à l'Échelle Locale-Interactions Soil Biosphere Atmosphere |
| AROME | Applications de la Recherche à l'Opérationnel à Méso-Echelle |
| WRF | Weather Research and Forecasting |
| SYNOP | surface synoptic observations |
| RUC | Rapid Updating Cycle |
| RMSE | Root Mean Square Error |
| MAE | Mean Absolute Error |
| POD | Probability of Detection |
| FAR | False Alarm Ratio |
| WMO | World Meteorological Organisation |
| DMN | Direction de la Météorologie Nationale (Moroccan Weather Departement) |
| CNRMSI | Centre National de Recherche Météorologique et Système d'Information |
| SMN | Service de Modélisation Numérique |

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
