# Peer review of "A Preliminary Impact Study of Wind on Assimilation and Forecast Systems into the One-Dimensional Fog Forecasting Model COBEL-ISBA over Morocco"

_atmosphere, doi:10.3390/atmos10100615_

Round 1

Reviewer 1 Report

General comment:

This paper presents an interesting study about the impact of the wind data on the fog forecast. The shown material promises to be suitable for publication in “Atmosphere”. The paper is well organized and written very clearly and legibly. However, I do have a few smaller comments and therefore, I recommend a minor revision.

The paper (in the discussion/conclusions) should emphasize:

1. What is the impact of the limited AMDAR interval (daytime) on the results, particularly taking into account the daily cycle of meteorological variables?

2. According to the Figs. 4 and 5, the lowest T-bias and T-RMSE are shown for lower levels (below 500 m) when NWP wind data are not included in the assimilation. Does this mean that the NWP model more mistakes in the lowermost part of the atmosphere? And why?

3. How does fog define from the 1D COBEL-ISBA model (Page 12, Section 5.3.3). I think that needs to be explained; using relative humidity, liquid water content, visibility or…

Detailed comments in text:

4. Page 1, line 19; please replace Wind observations into Assimilation of the wind observations… Observations and model are two divided things; something should link them together like assimilation….

5. I suggest expanding a little bit the discussion of research of the interaction between wind and fog like

Cuxart, J., Jimenez, M.A., 2012. Deep radiation fog in a wide Closed Valley: Deep Radiation Fog in a Wide Closed Valley: Study by Numerical Modeling and Remote Sensing. Pure Appl. Geophys. 169, 911–926.

Prtenjak, M. T., Klaić, M., Jeričević, A., Cuxart, J., 2018: The interaction of the downslope winds and fog formation over the Zagreb area. Atmos. Res. 214, 213-227.

Ryznar, E., 1977: Advection-radiation fog near Lake Michigan. Atmos. Environ., 11, 427–430.

6. Page 7, line 233; Please replace …zonal and meridonal… into …zonal and meridional components….

7. Page 8 line 259; please correct …experiment. 2016 period.

8. Page 9, line 292; please correct (oan) in …The beneficial impact oan analysis

9. Page 13; line 413; according to your results I think that you have more improvements in lower levels (below 500 m) (Figs. 4 & 5). It may be better to specify some height interval.

Reviewer 2 Report

This paper succinctly lays out a first set of experiments investigating the effects of assimilating wind observations into a single-column fog model.

My first main concern is that the discussion of the results is not adequately supported by the figures. Section 5.2 seems to argue that the addition of AMDAR data is beneficial, yet Figure 4 suggests that there is almost no difference between assimilating AMDAR (black) and not assimilating AMDAR (blue) on temperature background fields.  I was also surprised at how close the red and green lines are, given that green represents no wind assimilation at all, and red represents assimilating both AMDAR and SYNOP.  I believe the discussion of the results needs to be laid out more clearly and be consistent with the figures.

My second main concern is that Figure 5 does not appear to be consistent with Figure 4 (and similarly for Figures S2 and S3.)  In particular, at 50m, Figure 4 suggests that DA-NoWind has the best background field (smallest OmG RMSE).  However, Figure 5 shows that DA-NoWind has the worst RMSE, even at the first forecast hour.  How are these results reconciled?

Finally, in order to fully demonstrate the significance of the impacts of each type of data, the uncertainty in the results need to be quantified. That is, there should be confidence intervals on Figures 4-7 and S2-S3 to determine whether the “improvements” are statistically significant or not.

Minor comments:

Line 137: Should “expectation” be “expected variance” or “expected error”? Section 4 (methods): Clarify how the NWP data is used in DA-NoWind and in DA-Wind, emphasizing the differences between the two. Section 5.2 title and line 290: “analysis” and “background” mean two different things in data assimilation.Please don’t say “analysis background”, but clarify what is actually meant. Line 320-321 “which is found that are available in the upper levels and during some hours of the day” is very unclear, please reword. Line 324: what does “an important one” mean? Is this an important level? If so, why? Equations 4-5 should be presented earlier, since the bias and rmse are described prior to that section. In addition, please specify what i, nrefer to (is nthe total number of observations in time?) Line 356 “This confirms the detrimental impact of NWP data”: This is not supported by the curves referred to in the previous sentence. Line 387 the range of SS_FAR should be negative numbers.

There are several spelling/grammatical errors; I have only listed a few. Please carefully spell check.

Line 36 should be “Radiosondes are generally launched a few times per day, and do not directly measure the wind vectors.” Line 48-49 should be “the 3D NWP framework” and “the 1D framework”. Line 54 “particularly the radiation ones” – please reword. Line 64 should be “the 1D-Var method” Line 134 and elsewhere: “Thining” should be “Thinning” Line 267-268 “many previous researchs have shown” should be “previous research has shown” Line 278 should be “profiles” Line 320 “comparaison” should be “comparison”

Round 2

Reviewer 2 Report

This paper has been improved by the author in the revision, but there are several areas that still need some edits.  I recommend minor revisions.

General comments:

For OmG statistics, please explain exactly which observations are considered (what platforms?) I am still concerned about the consistency between Figures 4 and 5. Please explain why, in Figure 4b, DA-NoWind has errors similar to DA-Wind-NoNWP , but in Figure 5b DA-NoWind has larger errors comparable to DA-Wind and DA-NoAMDAR. The response to my original concern is only partly sufficient: DA-NoWind is significantly better than DA-Wind_NoAMDAR and DA-Wind in OmG statistics (Fig 4b), but in Fig 5b those three experiments are nearly identical.  Is this significant benefit entirely lost within an hour? Note that confidence intervals are themselves a statistical significance test. So, if confidence intervals overlap, then the differences are definitely not statistically significant (not “probably”, as stated on line 287 and in the response to my original review.)

Minor comments:

Line 19 change to “Assimilation of wind observations” Line 38: does “wind vector” mean “wind speed and direction”? Lines 162-163 change to “In fact, for radiation fog, the layers of the atmosphere in contact with the ground are the most important…” Line 176 don’t capitalize “the aircraft” Line 211: “a data denial tests” should be “data denial tests” Line 231 “observations” should be “observation” Lines 252-255 maybe change to “…upper levels where the B-matrix associated to the DA-Wind-NoNWP Experiment (SYNOP and AMDAR wind data are assimilated) has smaller standard deviation.” Lin 265 “impose” should be “imposes” Line 308 remove “statistics” Line 313: replace “on background” with “on the background fields” Line 317 replace “the numerical forecasting” with “forecasts” Figure 5: please move the legend on the bottom left so that it doesn’t overlap the data. Line 349 should be “On the other hand,…”
